bioengineering/psychology/biophysics

tactile feel, phonological impression, sensory evaluation, friction

**Author for correspondence:**
Yoshimune Nonomura
e-mail: nonoy@yz.yamagata-u.ac.jp

†These authors contributed equally to the study.

# Physical origin of a complicated tactile sensation: 'shittori feel'

Kana Kikegawa[1,†], Rieko Kuhara[1,†], Jinhwan Kwon[2], Maki Sakamoto[2], Reiichiro Tsuchiya[3], Noboru Nagatani[3] and Yoshimune Nonomura[1]

[1]Department of Biochemical Engineering, Graduate School of Science and Engineering, Yamagata University, 4-3-16 Jonan, Yonezawa 992-8510, Japan
[2]Department of Informatics, The University of Electro-Communications, 1-5-1 Chofugaoka, Chofu, Tokyo 182-8585, Japan
[3]Daito Kasei Kogyo, Co., Ltd, 1-6-28 Akagawa, Asahi, Osaka 535-0005, Japan

YN, 0000-0003-0461-124X

*Shittori* feel is defined as a texture that is moderately moisturized; however, many people experience 'shittori feel' when they touch a dry solid material containing little liquid. Here, *shittori* feel was evaluated for 12 materials. We found that the highest score of *shittori* feel was achieved by powders. Multiple regression analysis showed that *shittori* feel is a complex sense of moist and smooth feels. We analysed the relationship between the physical properties and the moist/smooth feels to show how subjects felt certain feels simultaneously. The moist and smooth feels are related to the surface roughness and friction characteristics of the materials. The moist and smooth feels can be perceived when the finger starts to move on the material surface and when the finger moves and rubs the material surface, respectively.

## 1. Background

When we evaluate tactile sense, a contradiction can arise between the definition of a term and the result of sensory evaluation. For example, the Japanese term 'shittori feel' is defined as 'a texture that is moderately moisturized' [1]. Although no word in English matches this word perfectly, the tactile texture is one of the most favourable textures for many Japanese and the most important for designing many products. In the field of cosmetics, *Shittori* feel is the most useful evaluation item for many formulators, because the feeling is the characteristic texture of healthy skin with high water holding capacity, desirable lotions and lipsticks [2–4]. In the case of textiles, this feeling is related to the favourability. Tanaka & Sukigara [5]

found that the stronger *shittori* feel was accompanied by either the warm or soft feel. However, this feel is quite complicated and involves some contradictions. Although this definition includes the word 'moisture', many people feel *shittori* feel when they touch a dry solid material containing little liquid.

In general, the result of sensory evaluation is analysed by tactile dimensions to understand the expression mechanism of tactile feel. Many tactile textures that arise when humans touch objects are described by a combination of simple tactile dimensions. Hollins *et al.* [6] evaluated the tactile texture of 17 objects, including paper, plastic and velvet, using the visual analogue scale (VAS) method and suggested that the tactile multidimensional space consists of three dimensions of smooth/rough, hard/soft and elastic. On the basis of the commonality found in a number of studies and known mechanisms for the perception of physical properties of textures, Okamoto *et al.* [7] concluded that the tactile textures are composed of at least five tactile dimensions: macro and fine roughness, hardness/ softness, coldness/warmness and friction (moistness/dryness and stickiness/slipperiness). In addition, they developed a method for investigating relationships among touch-related responses by constructing multilayered adjective models without assumptions regarding model structure such as adjective hierarchy or a definition of layers [8]. Takemura and co-workers [9] proposed a novel tactile evaluation system that can give tactile feedback from a sensor's output. The tactile sensation is classified from low order of tactile sensation (LTS, roughness, compliance, coldness and slipperiness) to high order of tactile sensation (HTS, embracingness, refreshingness, expensiveness), and also to preference. Here, LTS is correlated with physical measures including vibration, thermal property and friction.

The relationship between the aforementioned tactile dimensions and the physical stimulus applied to the skin, especially friction, is important to not only understand contradictions and complications in tactile sense but also to design cosmetics, clothes, cars and virtual reality systems. Smith *et al.* [10] examined the contribution of friction force to the subjective magnitude estimations of roughness. In tactile exploration, the root mean square of the tangential force rate can be an important determinant of subjective roughness. The relationship between friction and the tactile properties of woven and knitted fabrics was investigated [11]. A correlation between fabric friction and subjectively perceived touch properties was found for knitted fabrics but not for woven fabrics. Skedung *et al.* [12] examined the relationship between the measured friction coefficients and surface roughness to the perceived coarseness of the papers and found that both roughness and finger friction can be related to perceived coarseness. Tanaka and co-workers [13] focused on roughness perception of both coarse and fine textures of different materials (glass particle surfaces and sandpapers) and found that both spatial information (particle size) and temporal information (skin vibration) had a high correlation with subjective roughness ratings. The interaction forces between a human finger and the glass with different coatings were quantified and compared in terms of tactile friction coefficients [14]. Coated glass is characterized by high friction coefficient upon interaction with a human finger as well as significant hysteresis in the stroking directions (lower applied load and higher friction in the backward stroke).

We showed that the tactile feel when we touch the water on the glass surface is due to the periodic fluctuation of the frictional resistance [15,16]. When human hair was treated with silicone oil, more than 60% of the subjects felt their hair to be slippery and smooth [17]. Treating the subjects' hair with an oil-in-water emulsion after drying made them perceive a slippery feel because the surfactant reduced friction on the hair surface. These results suggested that both friction and thermal properties of the hair surface are important in controlling the tactile texture of human hair. Previous studies have shown that the *shittori* feel is caused by a small variation in the friction coefficient when the subject touches a smooth cold cloth or leather [18,19]. The effective characteristic values to understand the *shittori* feel of fabrics were the maximum value of heat flux, mean coefficient of friction, hysteresis of the shear force, shear stiffness and the compression energy.

We conducted a multifaceted analysis to clarify why humans experience a *shittori* feel even under dry conditions. We first evaluated the phonological impression of the term '*shittori* feel'. The term '*shittori* feel' was positioned on a haptic sensory space map. When a word is entered into the system, it is referenced by the phoneme database and auditory impression and is evaluated on the basis of 26 pairs of basic tactile evaluations [20]. Next, a *shittori* feel was evaluated for 12 materials containing powder, artificial leather, resin, metal, silicone rubber or cloth. We also evaluated 10 items with, for example, warm, cold, moist, and dry feels and analysed combinations of tactile dimensions that contribute to the *shittori* feel. In addition, we characterized the items' physical properties, including their coefficient of friction, surface roughness, surface energy, Young's modulus, thermal conductivity and water content. Because friction affects various dimensions, we imitated the physical conditions when humans touch objects. The surface of the object was rubbed with a contact probe that imitated the surface and mechanical properties and shape of human finger surfaces [21]. In addition, a sinusoidal friction evaluation device

was used to imitate the situation where the moving velocity is constantly changing when a human touches an object [22,23]. Finally, the relationship between sensory evaluation, tactile dimensions and physical properties was analysed to demonstrate the physical origin of *shittori* feel.

# 2. Material and methods

## 2.1. Materials

Table 1 shows the 12 solid materials. These materials were selected by the following method. We first searched objects described with the term '*shittori* feel' on the Internet. Next, three professional panelists determined 12 objects with various levels of *shittori* feel: one panelist was a researcher who has been engaged in tactile evaluations for more than 15 years, and the other two were students who have studied tactile sense.

## 2.2. Evaluation of phonological impression

The phonological impression of *shittori* feel was evaluated by a system that can convert a sound-symbolic word in Japanese into quantitative ratios in multiple tactile dimensions (26 pairs of adjectives) [20]. In this system, when a word that intuitively expresses a tactile sensation is input into the text field, information equivalent to evaluations against the 26 pairs of touch adjectives is obtained on the basis of an analysis of the sound of the word.

## 2.3 Sensory evaluation

Tactile evaluations were performed by 20 female students ranging in age from 20 to 29 years. The evaluations were performed in a quiet room at $25 \pm 1°C$ and $51 \pm 2\%$ in relative humidity after the subjects washed both hands with a commercial liquid hand soap and acclimatized for 20 min. The materials were evaluated in a random order to eliminate order effects, and the subjects touched the materials through a blackout curtain. The purpose of the tests was revealed to the subjects before the evaluations, and the subjects made the decision of whether to join the evaluation by themselves. All evaluations were conducted according to the principles expressed in the Declaration of Helsinki. The responsible party at Yamagata University confirmed that the ethics and safety of the present test were acceptable (approval number 29-12). Informed consent was obtained from all subjects.

When the subject touched solid materials on a table with the forefinger of their dominant hand and moved their finger to the left or right, they evaluated the tactile feel of the material for 25 s (figure 1). In the case of powders, the subjects picked them up with the thumb and forefinger of their dominant hand and rubbed their fingers together. After evaluation, the subjects answered four questions. Q1 and Q2 were as follows: 'When you touched the material, did you feel *shittori* feel?' and 'Why did you feel that?', respectively. Next, in Q3, the subjects evaluated 10 tactile sensations regarding the tactile dimensions: warm, cold, soft, hard, moist, dry, smooth, sticky, rough and slippery feels [7]. Finally, at Q4, the subjects freely described the tactile feel of the materials. In Q1 and Q3, the tactile sensation was evaluated on the basis of the VAS method. The VAS method is an evaluation method in which a subject marks on a horizontal straight line indicating the degree of a certain sensation. Subjects marked the most appropriate place on a 10 cm line with 'feel strong' or 'not at all' written at the ends [24]. The degree of sensation is quantified by the length between the mark and the end of 'not at all'. If the score of *shittori* feel was 7 or more, we asked timing when the *shittori* feel was felt. The choices were as follows: 1, 'The moment when you touch the material'; 2, 'while touching'; 3, 'the moment when you take your finger off'; 4, 'I do not know'.

## 2.4 Physical evaluation

The frictional force was evaluated using a sinusoidal motion friction evaluation system in which the contact probe slide on objects under sinusoidal motion was at $25 \pm 1°C$ and $51 \pm 2\%$ relative humidity [22,23]. Electronic supplementary material, figure S1 shows a photograph and an overview of the device. The sinusoidal motion was achieved by a scotch yoke mechanism that rotates an eccentric disc and makes the yoke reciprocate. The friction force applied to the contact probe $^1F_x$ and the forces applied to the material $^2F_x$ and $F_z$ were detected by strain gauge type load cells. The measurement range of the friction

**Table 1.** Image and chemical composition of 12 materials.

| material | image | chemical composition |
| --- | --- | --- |
| powder A |  | alkyl-silane-treated sericite, particle size = 10 μm, plateshaped, Daito Kasei Kogyo Co., Ltd, Osaka, Japan |
| powder B |  | N′-lauroyl-ʟ-lysine, particle size = 23 μm, plateshaped, Amihop LL, Ajinomoto Healthy Supply Co., Inc., Tokyo, Japan |
| artificial leather C |  | polyester/polyurethane, Toray Industries Inc., Tokyo, Japan |
| artificial leather D |  | polyester, Teijin Cordle Co., Ltd, Osaka, Japan |
| resin E |  | acrylic resin, Acrysunday Co., Ltd, Tokyo, Japan |
| resin F |  | PTFE, Naflon PTFE sheet TOMBO No. 9000, Nichias Co., Tokyo, Japan |
| metal G |  | aluminium, Tai Fung Trading Co., Tokyo, Japan |
| metal H |  | copper, Tai Fung Trading Co., Tokyo, Japan |
| silicone rubber I |  | silicone rubber on which surface structure of emery cloth #80 was transferred |
| silicone rubber J |  | silicone rubber on which surface structure of emery cloth #40 was transferred |
| cloth K |  | Japanese cashmere, Maruju, Fukuoka, Japan |
| cloth L |  | Swiss cashmere, Maruju, Fukuoka, Japan |

forces were as follows: $^{1}F_x = {}^{2}F_x = 0.06$–$9.9$ N and $F_z = 0.06$–$9.8$ N. By contrast, the detection limits of $^{1}F_x$, $^{2}F_x$, $F_z$ are $0.02$ N. In this study, $^{2}F_x$ and $F_z$ data obtained from the load cells on the sample stage were analysed because an inertial force was applied to the contact probe.

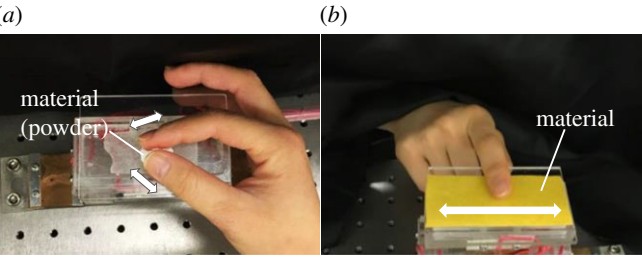

(a) (b)

**Figure 1.** Evaluation method of tactile feel for materials: (a) A̲ and B̲ and (b) C̲ − L̲.

The contact probe was a finger model made of urethane resin (electronic supplementary material, figure S1(c)) [21]. On the surface of this contact probe, grooves 0.15 mm deep are carved at 0.5 mm intervals to mimic fingerprints. Solid materials were adhered to an acrylic plate with double-sided tape (Nichiban Co., Ltd, Tokyo, Japan) to fix on the sample holder. The powder was sieved with a sieve opening of 0.5 mm and then dried for 12 h. Friction evaluations were carried out when 5 mg of the dried powder was applied to the artificial skin. The velocity $V$ during sinusoidal motion is described by the following equation

$$V = |D|\omega\cos\omega T,\tag{2.1}$$

where $D$, $\omega$ and $T$ are the moving width, angular velocity and time, respectively. The experimental conditions were as follows: $\omega = 2.1$ rad s$^{-1}$ (maximum velocity = 30 mm s$^{-1}$), vertical load = 0.98 N, $D = 30$ mm and number of cycle = 11.

The contact probe was in contact with the material for about 30 s prior to sliding. The coefficient of static friction depends on the conditions of friction. In particular, the contact time between the contact probe and the material before sliding can bring a significant effect on the static friction. Therefore, we confirmed that the static friction was almost the same even if the contact time was changed from 10 to 300 s.

To confirm the reproducibility, evaluations were conducted three times. The temporal profile of the angular velocity under this experimental condition is shown in electronic supplementary material, figure S2. Data for one round trip from the point where the contact probe changed direction for the first time were selected for quantitative analysis.

The friction data were analysed on the basis of the following five parameters: (i) static friction coefficient ($\mu_s$); (ii) kinetic friction coefficient ($\mu_k$); (iii) normalized delay time $\delta$ ($\delta = \Delta t/T_0$, where $\Delta t$ and $T_0$ are time lag between the velocity and the friction force and time for a cycle, respectively); (iv) $\mu_s - \mu_k$ (the difference between static and kinetic friction coefficients); and (v) the friction variation value (the standard deviation of the kinetic friction coefficient) (figure 2; electronic supplementary material, figure S3). Although $\delta$ is not common, we add this parameter for analysis, because we found that this parameter reflects some characteristic friction phenomena on soft matter surfaces in some previous studies [22,23].

The surface roughness of the material was measured using an LEXT OLS 4000 laser microscope (Olympus Co., Tokyo, Japan). The magnification of the objective lens was 5×, 10× or 20×. Three different locations on the material surface were observed to evaluate the surface roughness. In the case of powders, the surface was observed after 5 mg of powder was rubbed onto urethane artificial skin (Beaulax Co., Ltd, Saitama, Japan) using a sinusoidal motion friction evaluation system. The surface energy was cited from the literature [25–29]. The Young's modulus of powders, artificial leathers, silicone and cloths was measured using a Yawasa MEES-0512-1 (Tec Gihan Co., Ltd, Kyoto, Japan). The Young's modulus of resins and metals is cited from the literature [30]. The thermal conductivity of the artificial leather and cloths was measured using a KES-F7 Thermo Lab (Kato Tech Co., Ltd, Kyoto, Japan). That of powders, resins, metals and silicone rubber is cited from the literature [30,31]. The water content was measured using an MKC-510 Karl Fischer moisture meter (Kyoto Electronics Manufacturing Co., Ltd, Kyoto, Japan).

## 2.5. Statistical analysis

Multiple regression analysis was carried out using *shittori* feel and tactile dimensions as a target variable and explanatory variables, respectively. Statistical analyses were performed using the Excel version 14.0 (Microsoft Corp, Redmond, WA) and the SPSS 16.0 Base System software (IBM, New York, USA).

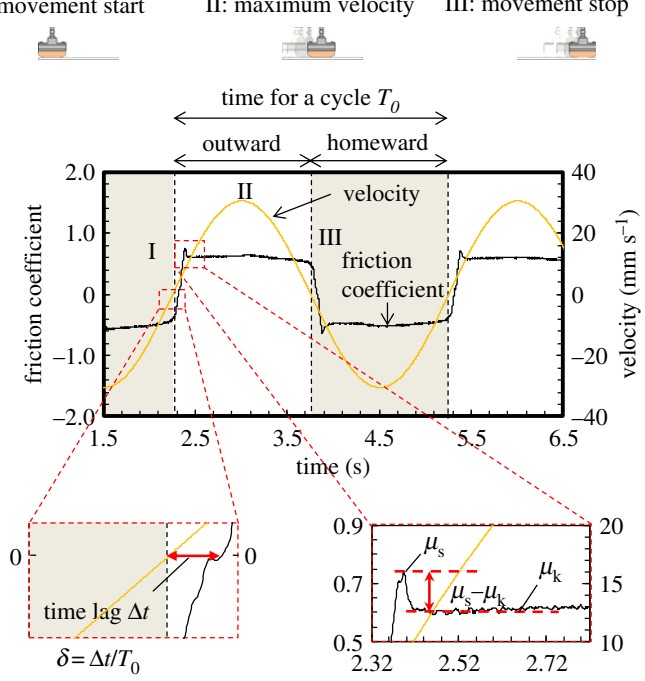

**Figure 2.** Typical friction profile and definition of parameters. The parameters $\mu_s$ and $\mu_k$ are static friction coefficient and kinetic friction coefficient, respectively.

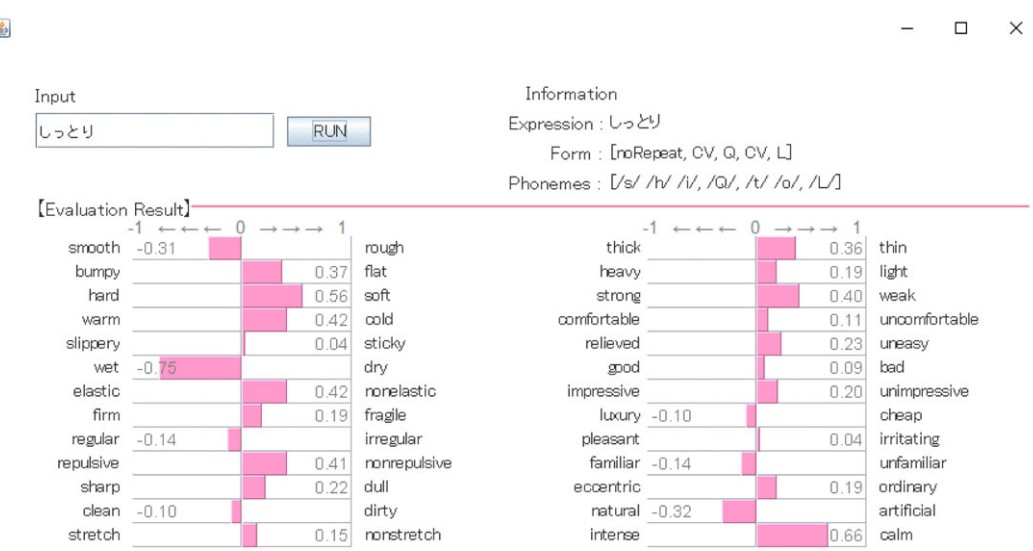

**Figure 3.** Output from the conversion system for *shittori*.

Student's *t*-test was conducted for each explanatory variable. The null hypothesis was rejected when the *p*-value was greater than 0.05 [32].

# 3. Results

## 3.1 Phonological impression

Figure 3 shows an output from the conversion system for *shittori*. The scores for 'moist (wet)', 'calm' and 'soft' were high: 0.75, 0.66 and 0.56, respectively. The evaluation scores of the tactile dimensions were as follows: moist = 0.75, soft = 0.56, cold = 0.42, smooth = 0.31 and sticky = 0.04. These evaluation results

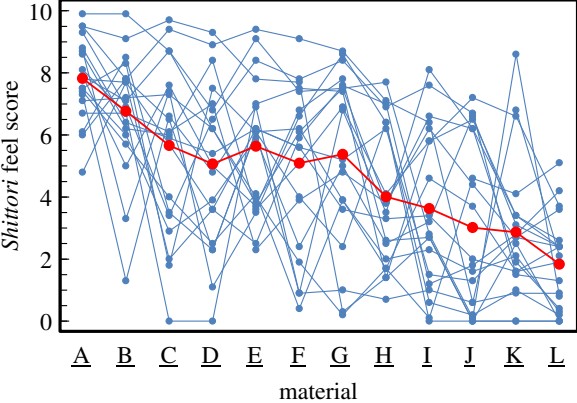

**Figure 4.** The *shittori* feel scores. Materials are powders (A̲, B̲), artificial leathers (C̲, D̲), resins (E̲, F̲), metals (G̲, H̲), silicone rubbers (I̲, J̲) and cloths (K̲, L̲). The score for all subjects and arithmetic means are plotted as small and large circles, respectively.

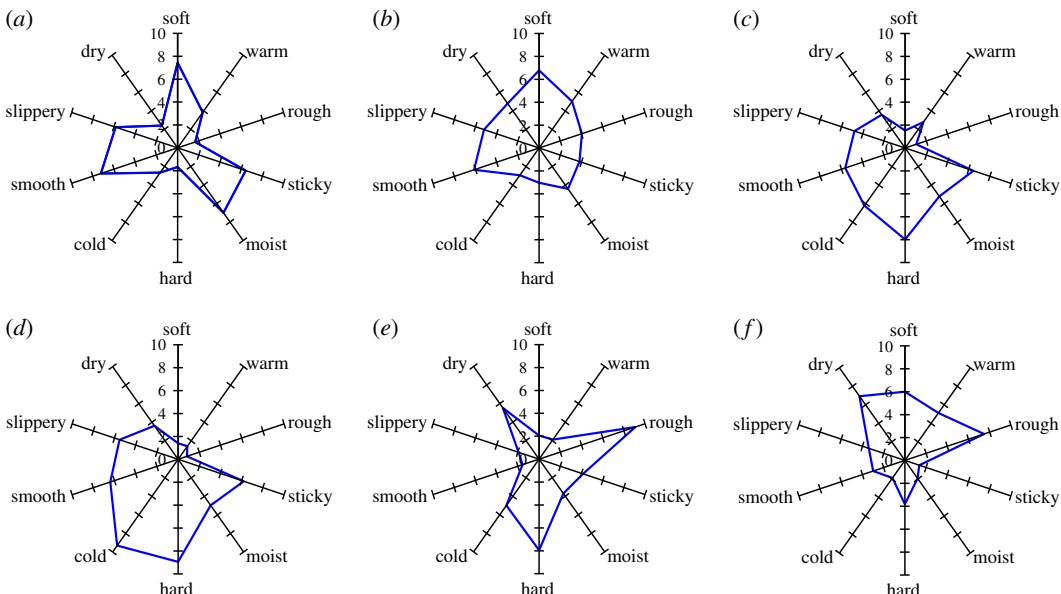

**Figure 5.** The score of tactile dimension for materials: (*a*) powder A̲, (*b*) artificial leather C̲, (*c*) resin E̲, (*d*) metal G̲, (*e*) silicone rubber J̲ and (*f*) cloth L̲.

suggest that the *shittori* feel is a complicated feel that cannot be explained with a single tactile dimension. Although the dominant factor is moist feel, one or more of soft, cold and smooth feels may also affect this sensation.

## 3.2. Sensory evaluation

The *shittori* feel score is shown in figure 4 and electronic supplementary material, table S1. Here, the results are expressed as arithmetic means and standard deviations. In figure 4, small and large circles are raw data and means of all subjects, respectively. The highest average scores were those for A̲ (7.8 ± 1.3, mean ± s.d.) and B̲ (6.8 ± 1.9), whereas the lowest scores were those for L̲ (1.8 ± 1.5) and K̲ (2.9 ± 2.2). A clear difference was observed between the groups A̲·B̲ and K̲·L̲. Figure 5 and electronic supplementary material, table S1 show the tactile score based on the tactile dimensions for each material. The powder with the highest *shittori* feel score strongly evoked soft, sticky, moist, smooth and slippery feels. In the comments of Q2 and Q4, the words concerning sticky or moist feel are appeared with relatively high probability for powders A̲ and B̲. On the other hand, the words related to dry feel were described for cloths K̲ and L̲ with the lowest score of *shittori* feel (electronic supplementary material, tables S2 and S3 and figure S4).

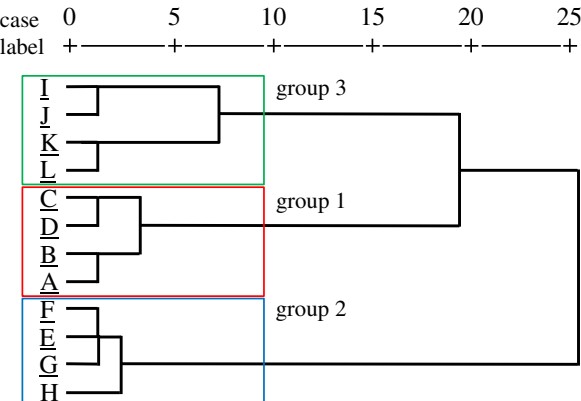

**Figure 6.** Cluster analysis for tactile dimension scores: powders (A̲, B̲), artificial leathers (C̲, D̲), resins (E̲, F̲), metals (G̲, H̲), silicone rubbers (I̲, J̲) and cloths (K̲, L̲).

By contrast, the scores of soft, warm, rough and dry feels were high when the subjects touched a cloth with the lowest *shittori* feel score. As mentioned in electronic supplementary material, table S1, many subjects felt the *shittori* feel while touching the materials: in the 64 cases with the *shittori* score of 7 or more, the number that the subjects felt 'at the moment when they touch the materials' was 19, while the number 'while touching' was 55.

Figure 6 shows the dendrogram obtained by cluster analysis of the tactile score of each material. The materials were classified into three groups according to tactile dimensions: group 1 = powders and artificial leathers (A̲, B̲, C̲ and D̲); group 2 = resins and metals (E̲, F̲, G̲ and H̲); and group 3 = silicone rubbers and cloths (I̲, J̲, K̲ and L̲) (figure 6). The powders with the highest *shittori* feel score and the fabrics with the lowest score were classified into different groups. Dimensions with a score greater than 5 were the soft, smooth and slippery feels for group 1.

## 3.3. Physical evaluations

Electronic supplementary material, table S4 shows the friction properties, surface roughness, surface energy, Young's modulus, thermal conductivity and the water content of the solid materials. We found a difference in the frictional parameters $\mu_s$ and $\mu_s - \mu_k$ between the powders A̲·B̲ with a high *shittori* feel score and the cloths K̲·L̲ with a low score. The static friction coefficients of A̲ with the highest *shittori* feel and L̲ with the lowest were 0.73 and 0.55, respectively; the difference between them was thus 0.18. The parameter $\mu_s - \mu_k$ was 0.18, 0.15, 0.05 and 0.09 for A̲, B̲, K̲ and L̲, respectively; thus, the value for the powders was more than twice as large as that for the fabrics. These results suggest that the frictional resistance is strong at the start of sliding in the case of a material with a strong *shittori* feel.

Surface roughness was especially large for C̲·K̲·L̲, whose surface was covered with fur, and I̲·J̲, whose surface was uneven: $R_a$ = 44.7, 40.6, 116, 105 and 93.9 μm for C̲, I̲, J̲, K̲ and L̲, respectively. The other materials were several to 10 μm. The surface energy, Young's modulus and thermal conductivity showed large values characteristic of metals. However, no relationship was found between these values and the score of *shittori* feel. The water content was also high, 7.38 and 6.91%, for fabrics K̲ and L̲, respectively, although no clear relationship with the fabrics' *shittori* feel was found.

## 3.4. Tactile dimensions which affect *shittori* feel

The *shittori* feel is correlated with many tactile dimensions. The correlation coefficients associated with moist, sticky, smooth, slippery, dry and rough feels were 0.707, 0.477, 0.391, 0.290, −0.558 and −0.443, respectively. A regression analysis was achieved using *shittori* feel as a target variable and a combination of moist feel and other tactile dimensions as explanatory variables. Among some regression formulas satisfying conditions of regression analysis, the below equation shows the highest correlation coefficient obtained with the sensory evaluation data

$$Y^{\text{Shittori}} = 0.943 + 0.659 X^{\text{Moist}} + 0.195 X^{\text{Smooth}}, \tag{3.1}$$

where $Y^{\text{Shittori}}$, $X^{\text{Moist}}$ and $X^{\text{Smooth}}$ are the score of *shittori*, moist and smooth feels, respectively.

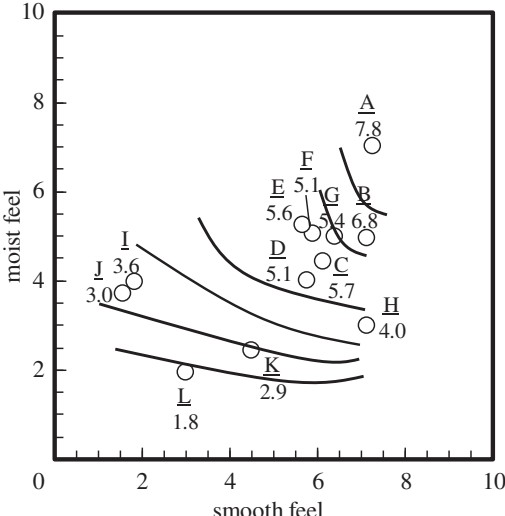

**Figure 7.** Relationship between *shittori*, moist and smooth feels. The numerical values represent the *shittori* feel score.

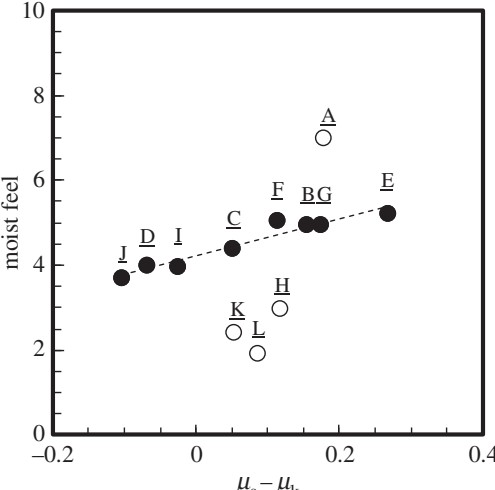

**Figure 8.** Effect of $\mu_s - \mu_k$ on the moist feel score.

The explanatory variable was moist and smooth feels. The multiple correlation coefficient $R$ for equation (3.1) and the correlation coefficient between explanatory variables (scores of moist and smooth feels) were 0.740 and 0.258, respectively. The significance probability of each explanatory variable $p$-value was less than 0.05 for moist and smooth feels. Figure 7 shows the relationship between *shittori*, moist and smooth feels. The numerical values in the graph are a score representing the strength of *shittori* feel. In cases where the score of both moist and smooth feels was higher, the *shittori* feel became stronger. To perceive the *shittori* feel, a complex tactile sensation of moist and smooth is necessary.

## 3.5. Physical origin of *shittori* feel

We revealed the physical origin of moist and smooth feels, which are related to the *shittori* feel. In particular, we wondered why many people perceived a moist feel that is generally thought to be caused by wet conditions even though they touched a dry solid material containing little liquid. Significant correlation indicating that water content and thermal conductivity lead to the enhancement of the moist feel was not observed. However, we found an interesting trend in the friction profile: larger values of $\mu_s - \mu_k$ are associated with stronger moist feels. Figure 8 shows the relationship

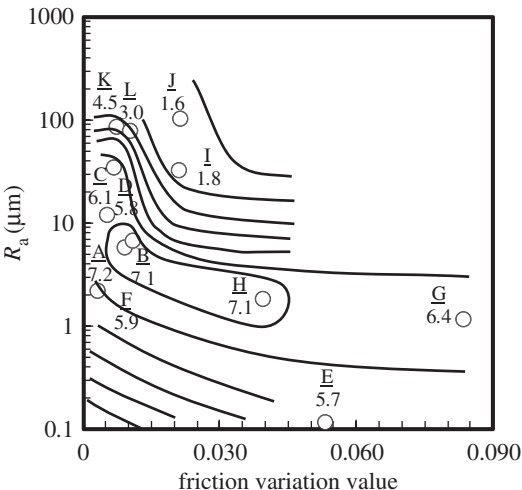

**Figure 9.** Relationship between smooth feel, $R_a$ (an objective lens with 10× magnification was used) and the friction variation value. The numerical values are the smooth feel scores.

between moist feel and $\mu_s - \mu_k$. For eight materials except $\underline{A}$, $\underline{H}$, $\underline{K}$ and $\underline{L}$, a linear relationship between these sensory and physical parameters was observed

$$X^{\text{Moist}} = 4.22 + 4.37(\mu_s - \mu_k). \tag{3.2}$$

The explanatory variable was $\mu_s - \mu_k$. The correlation coefficient $R$ for equation (3.2) was 0.918. The significance probability of explanatory variable $p$-values was less than 0.05 for $\mu_s - \mu_k$. However, some exceptions that did not follow this linear relationship were found. For example, metal $\underline{H}$ and fibres $\underline{K}$ and $\underline{L}$ showed moist feels lower than those expected on the basis of $\mu_s - \mu_k$. These smaller-than-expected moist feels might be caused by inhibition by a strong dry feel. In response to Q2, 20, 50 and 35% of subjects used words related to the dry feel to describe metal $\underline{H}$ and fibres $\underline{K}$ and $\underline{L}$. These results suggest that many subjects consciously felt dry feels when they touched these objects. Under such circumstances, subjects are apparently reluctant to give a high evaluation score to moist feel, which is the opposite sense to dry feel. By contrast, powder $\underline{A}$ showed a moist feel higher than that expected on the basis of $\mu_s - \mu_k$. We do not have a clear explanation for this phenomenon. At the present time, no psychological or physical parameters that can rationally explain the high moist feel of powder $\underline{A}$ have been found.

Smooth feel showed a strong negative correlation with surface roughness $R_a$ when a 10× objective lens was used (correlation coefficient = −0.769). Previous studies have suggested that a smooth feel is perceived when the variation of the frictional force with respect to the average frictional force is small [33–35]. Figure 9 shows the relationship between smooth feel, surface roughness and friction variation value. Numerical values in the graph are the smooth feel scores. In the diagram, we found two distinct regions: the first region was composed of powders and artificial leathers, and the second was composed of resins and metals. Some characteristics were observed for each of the two groups: the first group exhibited a small variation in frictional resistance, and the surface roughness was several to several tens of micrometres ($\underline{A}$, $\underline{B}$, $\underline{C}$, $\underline{D}$). In the second group, the surface roughness was less than a few micrometres ($\underline{G}$, $\underline{H}$). These results suggest that there are two types of perceptual processes in the sense of smooth feel.

Figure 10 shows the perceived path of *shittori* feel. The *shittori* feel is perceived through the two senses of moist and smooth feels. In addition, these two senses relate to the surface roughness and friction characteristics of the materials.

## 4. Discussion

Why do subjects feel a *shittori* feel even under dry conditions? As mentioned in figure 8, the score of the moist feel became larger with the increase in $\mu_s - \mu_k$. The strong resistance applied to the skin surface at the moment of sliding is related to moist feel. This result is consistent with several previous studies. Some researchers have suggested that frictional resistance increases if there is a small amount of water on the skin surface. This increase in friction is caused by the capillary adhesion at liquid bridges between

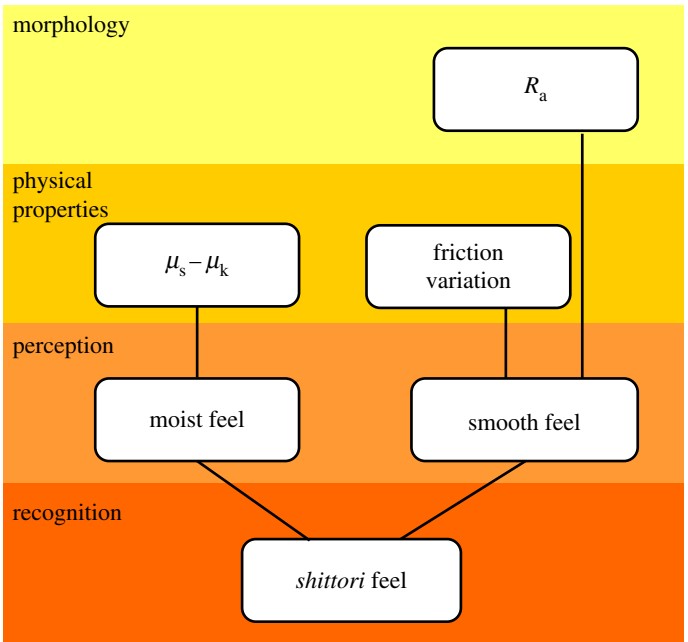

**Figure 10.** Relationship diagram between *shittori*, tactile factors and physical factors.

the skin and the object [36]. The macroscopic meniscus creates capillary attraction force, $F_{at}$ [37]. The capillary attraction force between surfaces due to a liquid bridge is determined by the surface tension of water, the size of contact region, the radius of the liquid meniscus and the average thickness of the thin liquid film. Although powders A and B, resin E and metal G were dry materials containing almost no water, many subjects can remember the resistant force of a substance containing a little water on the basis of its large friction difference $\mu_s - \mu_k$.

The hypothesis based on the capillary attraction force may be too simple to explain the phenomena that occur on the actual skin surface. For example, the extent of moisture accumulation in the contact region is associated with an increase in the friction [38]. The morphology of the sliding surface in the powder bed is complex: during the sliding process, the powder is packed into the groove of the fingerprint, which lowers the friction clearly [39]. The uneven surface of the paper absorbs sweat and changes the friction characteristics drastically [40]. Dussaud *et al*. [41] measured flows within the fine channels on the human skin surface and demonstrated that this network can act as a microfluidic device to allow the surface transport of liquids.

We consider a mechanism where different tactile feels of moist and smooth are recognized when a subject touches a material with *shittori* feel. The variation from the moment when the finger touches the object to the end of the finger movement might enable the make perception of complex tactile sensations. The moist feel is perceived the instant the finger starts to move on the material surface, whereas the smooth feel is felt when the finger moves as rubbing the material. First, the moist feel was related to the difference between static and kinetic friction coefficients, suggesting that the moist feel is perceived at the moment when the finger starts to slide because static friction is a force generated when an object begins to move. Next, smooth feel is perceived as the finger rubs the materials, because humans recognize roughness and smoothness through vibrations and deformation of the skin caused by rubbing the surface of the material with their fingers [42–44].

# 5. Conclusion

We found that *shittori* feel, which is defined as 'a texture that is moderately moisturized', is a complex sense of moist and smooth feels. The moist and smooth feels relate to the surface roughness and friction characteristics of the material. The moist feel is perceived the instant the finger starts to move on the material surface, whereas the smooth feel is perceived in the process when the finger moves as rubbing the material.

These findings are important for sharing and using information on tactile sense among people all over the world. It is difficult to understand each other among people with different backgrounds, because the sense strongly depends on language and culture. There is no doubt that the *shittori* feel is

an important factor in designing clothes, cosmetics and food in the Japanese market. However, people with different backgrounds did not understand the feeling, so it was a major obstacle in the global company that is developing business in Japan.

The model that explains the *shittori* feel by the tactile dimensions and the common parameters such as the friction coefficient will make it possible to design and evaluate products based on this sense all over the world. Moreover, it was confirmed that moist and smooth feels can be explained by friction and surface shape. Interestingly, the relationship between these sensory factors and physical factors was not a simple linear relationship. One of the reasons is that nonlinear mechanical response occurs on soft matter surfaces such as skin. In future, development of a model that accurately describes such a nonlinear response is required.

Ethics. The study was conducted under the permission of the ethical committee of the responsible party at Yamagata University (approval number 29-12).

Data accessibility. The datasets supporting this article have been uploaded as part of the electronic supplementary material.

Authors' contributions. K.K. and R.K. carried out the experimental work, performed the data analysis, carried out the statistical analyses, designed the study and drafted the manuscript; J.K. and M.S. carried out the evaluation of phonological impression; R.T. and N.N. prepared powder materials and carried out the evaluation of water content; Y.N. participated in the design of the study, coordinated the study and helped draft the manuscript; all authors gave final approval for publication.

Competing interests. We declare we have no competing interests.

Funding. This work was supported by Grant-in-Aids for Scientific Research on Innovative Area (no. 16H01661) and for Scientific Research (B) (no. 18H01402) from the Ministry of Education, Culture, Sports, Science and Technology, Japan (MEXT).

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
