## [Reviewer comments · Royal Society Open Science]

Review History

RSOS-190039.R0 (Original submission)

Review form: Reviewer 1

Is the manuscript scientifically sound in its present form?

No

Are the interpretations and conclusions justified by the results?

No

Is the language acceptable?

Yes

Is it clear how to access all supporting data?

Yes

Do you have any ethical concerns with this paper?

No

Have you any concerns about statistical analyses in this paper?

No

Recommendation?

Reject

Comments to the Author(s)

The authors describe the tactile perception of a range of materials and rationalise the data mainly on the basis of corresponding measurements of their friction using a urethane resin probe. In particular, they argue that the difference between the static and dynamic values of the friction are an important contributory factor. Some of the differences are negative so that the meaning is not clear. The concept of a static friction is nebulous and is often only associated with extended contact times prior to sliding. To be convincing, the effect of this delay time should have been examined. In any case, the frictional characteristics of a finger pad are complex and cannot be satisfactorily mimicked by a probe. This is due to the extent of moisture accumulation in the contact region that is associated with an increase in the friction [e.g. Dzidek et al., 2017. Why pens have rubbery grips. Proc. Nat. Acad. Sci., 114, pp.10864-10869.] This so called occlusion effect is absent on rough surfaces and the friction reduces with sliding time for absorbent surfaces such as paper and probably some fabrics [Adams et al., 2013. Finger pad friction and its role in grip and touch. J. Roy. Soc. Interf., 10, p.20120467.] It will also be seen in this latter reference that the adhesion, due for example to capillary forces, is negligibly small compared to the applied normal forces. Consequently, an explanation based on capillary adhesion (eq. 4) is not correct. I suggest that the authors review the extensive literature on finger pad friction in order to provide a more convincing explanation of their data.

Review form: Reviewer 2

Is the manuscript scientifically sound in its present form?

Yes

Are the interpretations and conclusions justified by the results?

Yes

Is the language acceptable?

Yes

Is it clear how to access all supporting data?

Yes

Do you have any ethical concerns with this paper?

No

Have you any concerns about statistical analyses in this paper?

No

Recommendation?

Major revision is needed (please make suggestions in comments)

Comments to the Author(s)

“Shittori feel” is an interesting topic.

The authors conducted the multiple regression analyses of different materials.

The authors analyzed the relationship between the physical properties and the moist/smooth feels to show how subjects felt certain feels simultaneously.

However, the following problems must be paid attention to.

1. Suggest that you add these diagrams and tables (figure S1, figure S2, figure S3, table S1...) to the proper place of the text so that the reader can understand them better.
2. For similar lists, recommend using tables.
3. The references in the past five years account for less than one sixth. Hope to refer to more recent articles.

Review form: Reviewer 3

Is the manuscript scientifically sound in its present form?

Yes

Are the interpretations and conclusions justified by the results?

Yes

Is the language acceptable?

Yes

Is it clear how to access all supporting data?

No

Do you have any ethical concerns with this paper?

No

Have you any concerns about statistical analyses in this paper?

Yes

Recommendation?

Accept with minor revision (please list in comments)

Comments to the Author(s)

The paper is clearly written, the figures are easy to read and the information flows nicely. The authors followed a systematic and comprehensive approach to understanding what contributes to the shittori feel, by comparing how perceptual evaluations of material properties and materials properties relate to each other and to the overall assessment of the shittori effet.

Specific comments:

- The variations in responses amongst participants are not shown nor discussed. It seems the score and values are the means across all participants but that is not clearly stated. I would really like to see the distribution of results across participants and a detailed explanation on how exactly the scores were computed.
- The images, the equipment used for measurements and the sample preparation are very detailed, however, I don't think the raw data of individual responses to all the questions asked in the experiment is accessible and I think the journal requires it.
- It is unclear if the responses to Q2, Q4 and follow-up questions about when the effect was felt

- were analyzed or discussed in the paper. It would be good to add that or make it more explicit.
- I would suggest adding a bit more detail on the importance of the "Shittori" feel.
 - Could you please specify the type/model of the load cells used to measure the frictions forces?
 - I would remove the variable "delay time". It's not super common and it doesn't add to your analysis.
 - I would probably remove the capillary force equation since you just state it without going further. It would also make the text flow more instead of breaking it up as it is currently.
 - Did the participants all know the shittori effect? Were they provided with a definition at the beginning of the experiments to ensure consistency in reporting across all participants?

Decision letter (RSOS-190039.R0)

24-Apr-2019

Dear Dr Nonomura,

The editors assigned to your paper ("Physical origin of a complicated tactile sensation: "Shittori feel") have now received comments from reviewers. We would like you to revise your paper in accordance with the referee and Associate Editor suggestions which can be found below (not including confidential reports to the Editor). Please note this decision does not guarantee eventual acceptance.

Please submit a copy of your revised paper before 17-May-2019. Please note that the revision deadline will expire at 00.00am on this date. If we do not hear from you within this time then it will be assumed that the paper has been withdrawn. In exceptional circumstances, extensions may be possible if agreed with the Editorial Office in advance. We do not allow multiple rounds of revision so we urge you to make every effort to fully address all of the comments at this stage. If deemed necessary by the Editors, your manuscript will be sent back to one or more of the original reviewers for assessment. If the original reviewers are not available, we may invite new reviewers.

If your study uses humans or animals please include details of the ethical approval received, including the name of the committee that granted approval. For human studies please also detail

whether informed consent was obtained. For field studies on animals please include details of all permissions, licences and/or approvals granted to carry out the fieldwork.

- Data accessibility

If you wish to submit your supporting data or code to Dryad (<http://datadryad.org/>), or modify your current submission to dryad, please use the following link:
<http://datadryad.org/submit?journalID=RSOS&manu=RSOS-190039>

- Competing interests

- Authors' contributions

- Acknowledgements

- Funding statement

Kind regards,
Andrew Dunn
Royal Society Open Science Editorial Office

on behalf of Dr Derek Abbott (Associate Editor) and R. Kerry Rowe (Subject Editor)
openscience@royalsociety.org

Comments to Author:

Reviewers' Comments to Author:

Reviewer: 1

Comments to the Author(s)

The authors describe the tactile perception of a range of materials and rationalise the data mainly on the basis of corresponding measurements of their friction using a urethane resin probe. In particular, they argue that the difference between the static and dynamic values of the friction are an important contributory factor. Some of the differences are negative so that the meaning is not clear. The concept of a static friction is nebulous and is often only associated with extended contact times prior to sliding. To be convincing, the effect of this delay time should have been examined. In any case, the frictional characteristics of a finger pad are complex and cannot be satisfactorily mimicked by a probe. This is due to the extent of moisture accumulation in the contact region that is associated with an increase in the friction [e.g. Dzidek et al., 2017. Why pens have rubbery grips. *Proc. Nat. Acad. Sci.*, 114, pp.10864-10869.] This so called occlusion effect is absent on rough surfaces and the friction reduces with sliding time for absorbent surfaces such as paper and probably some fabrics [Adams et al., 2013. Finger pad friction and its role in grip and touch. *J. Roy. Soc. Interf.*, 10, p.20120467.] It will also be seen in this latter reference that the adhesion, due for example to capillary forces, is negligibly small compared to the applied normal forces. Consequently, an explanation based on capillary adhesion (eq. 4) is not correct. I suggest that the authors review the extensive literature on finger pad friction in order to provide a more convincing explanation of their data.

Reviewer: 2

Comments to the Author(s)

“Shittori feel” is an interesting topic.

The authors conducted the multiple regression analyses of different materials.

The authors analyzed the relationship between the physical properties and the moist/smooth feels to show how subjects felt certain feels simultaneously.

However, the following problems must be paid attention to.

1. Suggest that you add these diagrams and tables (figure S1, figure S2, figure S3, table S1...) to the proper place of the text so that the reader can understand them better.
2. For similar lists, recommend using tables.
3. The references in the past five years account for less than one sixth. Hope to refer to more recent articles.

Reviewer: 3

Comments to the Author(s)

The paper is clearly written, the figures are easy to read and the information flows nicely. The authors followed a systematic and comprehensive approach to understanding what contributes to the shittori feel, by comparing how perceptual evaluations of material properties and materials properties relate to each other and to the overall assessment of the shittori effect.

Specific comments:

- The variations in responses amongst participants are not shown nor discussed. It seems the score and values are the means across all participants but that is not clearly stated. I would really like to see the distribution of results across participants and a detailed explanation on how exactly the scores were computed.
- The images, the equipment used for measurements and the sample preparation are very detailed, however, I don't think the raw data of individual responses to all the questions asked in the experiment is accessible and I think the journal requires it.
- It is unclear if the responses to Q2, Q4 and follow-up questions about when the effect was felt were analyzed or discussed in the paper. It would be good to add that or make it more explicit.
- I would suggest adding a bit more detail on the importance of the "Shittori" feel.
- Could you please specify the type/model of the load cells used to measure the frictions forces?
- I would remove the variable "delay time". It's not super common and it doesn't add to your analysis.
- I would probably remove the capillary force equation since you just state it without going further. It would also make the text flow more instead of breaking it up as it is currently.
- Did the participants all know the shittori effect? Were they provided with a definition at the beginning of the experiments to ensure consistency in reporting across all participants?

Author's Response to Decision Letter for (RSOS-190039.R0)

See Appendix A.

Decision letter (RSOS-190039.R1)

10-Jun-2019

Dear Dr Nonomura,

I am pleased to inform you that your manuscript entitled "Physical origin of a complicated tactile sensation: "Shittori feel"" is now accepted for publication in Royal Society Open Science.

on behalf of Dr Derek Abbott (Associate Editor) and R. Kerry Rowe (Subject Editor)
openscience@royalsociety.org

Appendix A

Yamagata University
Department of Biochemical Engineering,
Graduate School of Science and Engineering
4-3-16, Jonan, Yonezawa, Yamagata 992-8510, Japan
Phone: +81-238-26-3164 Fax: +81-238-26-3414
E-mail: nonoy@yz.yamagata-u.ac.jp

May 25, 2019

Dear Editors,

This submission contains the revised version of our manuscript (ID RSOS-190039) titled ‘Physical origin of a complicated tactile sensation: “*Shittori* feel”’. We thank you and reviewers for giving us some important suggestions. We changed the manuscript according to reviewer’s advices. We believe that the revised paper is suitable to be published in *Royal Society Open Science*.

Reviewer 1:

I would like to appreciate your suggestive comments. In the present paper, we have revised these points.

•“Some of the differences are negative so that the meaning is not clear. The concept of a static friction is nebulous and is often only associated with extended contact times prior to sliding. To be convincing, the effect of this delay time should have been examined.”

The contact probe was in contact with the material for 30 seconds prior to sliding. As the reviewer mentioned, the static friction coefficient depends on the conditions of sliding. In particular, the contact time between the contact probe and materials before sliding can bring a significant effect on the static friction. Therefore, we confirmed that the static and dynamic friction coefficients were almost the same even if the contact time was changed from 30-300 seconds (section 2.4 p. 10 line 22- p.11 line 3). However, it was difficult to show the effect of delay time, because the time was only 0.05-0.06 seconds and was too short to check the effect precisely. We will conduct systematic analysis to show the effect of the delay time after developing a more precise evaluation method.

Figure Effects of contact time on static friction coefficient.

Figure Effects of contact time on dynamic friction coefficient.

•“In any case, the frictional characteristics of a finger pad are complex and cannot be satisfactorily mimicked by a probe. This is due to the extent of moisture accumulation in the contact region that is associated with an increase in the friction [e.g. Dzidek *et al.*, 2017. *Why pens have rubbery grips. Proc. Nat. Acad. Sci.*, 114, pp.10864-10869.] This so called occlusion effect is absent on rough surfaces and the friction reduces with sliding time for absorbent surfaces such as paper and probably some fabrics [Adams *et al.*, 2013. *Finger pad friction and its role in grip and touch. J. Roy. Soc. Interf.*, 10, p.20120467.] It will also be seen in this latter reference that the adhesion, due for example to capillary forces, is negligibly small compared to the applied normal forces. Consequently, an explanation based on capillary adhesion (eq. 4) is not correct. I suggest that the authors review the extensive literature on finger pad friction in order to provide a more convincing explanation of their data.”

The hypothesis based on the capillary adhesion force may be too simple to explain the phenomena that occur on the actual skin surface. In future, we will propose more suitable model of interfacial phenomena on human skin. For example, as the reviewer mentioned, the extent of moisture accumulation in the contact region is associated with an increase in the friction [38]. The morphology of the sliding surface in the powder bed is complex: During the sliding process the powder is packed into the groove of the fingerprint, which lowers the friction significantly [39]. The uneven surface of the paper absorbs sweat and changes the friction characteristics drastically [40]. Dussaud *et al.* measured flows within the fine channels on the human skin surface and demonstrated that this network can act as a microfluidic device to allow the surface transport of liquids. [41]. The description was added in page 23 lines 4-12.

Reviewer 2:

We thank you for appreciating our manuscript very much.

•“Suggest that you add these diagrams and tables (figure S1, figure S2, figure S3, table S1 ...) to the proper place of the text so that the reader can understand them better.”

In the revised paper, Figure S1, S2, and S6 are added in the main text (Figure 1, 2, Table1). Others are remained in SI because they are figures of instructions and data.

•“For similar lists, recommend using tables.”

The images and chemical composition of 12 materials, which were described in Figure S1 and section 2.1. in the previous version, are shown in Table 1.

•“The references in the past five years account for less than one sixth. Hope to refer to more recent articles.”

Some recent papers on modeling of tactile texture and tribology of human skin were added as references [8], [9], [13], [14], [38], and [44].

Reviewer 3:

I would like to appreciate your suggestive comments. In the present paper, we have revised these points.

•“*The variations in responses amongst participants are not shown nor discussed. It seems the score and values are the means across all participants but that is not clearly stated. I would really like to see the distribution of results across participants and a detailed explanation on how exactly the scores were computed.*”

In the revised paper, *shittori* score of all subjects, means and standard deviations are described in Figure 2 and Table S1 to show the distribution of results across participants. The means are arithmetic averages. The descriptions are added in page 14 lines 2-12.

•“*The images, the equipment used for measurements and the sample preparation are very detailed, however, I don't think the raw data of individual responses to all the questions asked in the experiment is accessible and I think the journal requires it.*”

The raw data of individual responses to all the questions are added in Table S1-3. The descriptions were added in the section 3.2. In the comments of Q2 and 4, the words concerning sticky or moist feel are appeared with relatively high probability for powders A and B. On the other hand, the words related to dry feel were described for cloths K and L with the lowest score of *shittori* feel (Table S2,3 and Fig. S4).

•“*It is unclear if the responses to Q2, Q4 and follow-up questions about when the effect was felt were analyzed or discussed in the paper. It would be good to add that or make it more explicit.*”

The comments and appearance rate of the words on each tactile dimension in the subject's comment were added in Table S2, S3 and Figure S4. In the comments of the subjects of Q2 and 4, the words concerning sticky or moist feel appeared with relatively high probability for powders A and B. On the other hand, the words related to dry feel were described for cloths K and L with the lowest score of *shittori* feel. These descriptions were added in the section 3.2 page 14 lines 9-12.

•“*I would suggest adding a bit more detail on the importance of the “Shittori” feel.*”

In the field of cosmetics, *Shittori* feel is the most important evaluation item for many formulators, because the feeling is the characteristic texture of healthy skin with high water

holding capacity, desirable lotions and lipsticks [2-4]. In the case of textile, this feel is related to the favorability. Tanaka and Sukigara found that the stronger *shittori* feel was accompanied with either the warm or soft [5]. These references were added in introduction (page 2 lines 18-23).

•“*Could you please specify the type/model of the load cells used to measure the frictions forces?*”

The friction force applied to the contact probe 1F_x and the forces applied to the material 2F_x and F_z were detected by strain gauge type load cells, which are order-made. The measurement range of the friction forces were as follows: ${}^1F_x = {}^2F_x = 0.06\text{--}9.9$ N and $F_z = 0.06\text{--}9.8$ N. In contrast, the detection limits of ${}^1F_x, {}^2F_x, F_z = 0.02$ N. These descriptions are added in the section 2.4 page 10 lines 5-8.

•“*I would remove the variable “delay time”. It’s not super common and it doesn’t add to your analysis.*”

Although δ is not common, we add this parameter for the analysis, because we found that this parameter reflects some characteristic friction phenomena on soft matter surfaces in some previous studies [22, 23]. This description was added in the section 2.4 (page 11 lines 13-15).

•“*I would probably remove the capillary force equation since you just state it without going further. It would also make the text flow more instead of breaking it up as it is currently.*”

As you mentioned, the equation (4) was deleted in the revised paper.

•“*Did the participants all know the shittori effect? Were they provided with a definition at the beginning of the experiments to ensure consistency in reporting across all participants?*”
Shittori feel is a very common word in Japanese, and listed as almost the same meanings in typical dictionaries. We did not explain the meaning before evaluation, because all subjects were native Japanese speakers.

I thank the referees for their comments and hope the manuscript is now suitable for publication in *Royal Society Open Science*.

Yours sincerely,
Yoshimune Nonomura
Department of Biochemical Engineering,
Graduate School of Science and Engineering,
Yamagata University,
4-3-16, Jonan, Yonezawa, Japan
nonoy@yz.yamagata-u.ac.jp